# Research

health and disease and epidemiology, theoretical biology

mathematical epidemiology, reproductive number, exponential growth rate, speed and strength

**Author for correspondence:**
Jonathan Dushoff
e-mail: dushoff@mcmaster.ca

# Speed and strength of an epidemic intervention

Jonathan Dushoff[1,2,3] and Sang Woo Park[4]

[1]Department of Biology, [2]Department of Mathematics and Statistics, and [3]M. G. DeGroote Institute for Infectious Disease Research, McMaster University, Hamilton, Ontario, Canada
[4]Department of Ecology and Evolutionary Biology, Princeton University, Princeton, NJ, USA

 JD, 0000-0003-0506-4794; SWP, 0000-0003-2202-3361

An epidemic can be characterized by its strength (i.e., the reproductive number $\mathcal{R}$) and speed (i.e., the exponential growth rate $r$). Disease modellers have historically placed much more emphasis on strength, in part because the effectiveness of an intervention strategy is typically evaluated on this scale. Here, we develop a mathematical framework for the classic, strength-based paradigm and show that there is a dual speed-based paradigm which can provide complementary insights. In particular, we note that $r = 0$ is a threshold for disease spread, just like $\mathcal{R} = 1$ [1], and show that we can measure the strength and speed of an intervention on the same scale as the strength and speed of an epidemic, respectively. We argue that, while the strength-based paradigm provides the clearest insight into certain questions, the speed-based paradigm provides the clearest view in other cases. As an example, we show that evaluating the prospects of 'test-and-treat' interventions against the human immunodeficiency virus (HIV) can be done more clearly on the speed than strength scale, given uncertainty in the proportion of HIV spread that happens early in the course of infection. We also discuss evaluating the effects of the importance of pre-symptomatic transmission of the SARS-CoV-2 virus. We suggest that disease modellers should avoid over-emphasizing the reproductive number at the expense of the exponential growth rate, but instead look at these as complementary measures.

## 1. Introduction

An epidemic can be described by its *strength* and *speed*. The strength of an epidemic is characterized by the reproductive number $\mathcal{R}$, which measures *how many* new infections are caused by a typical infected individual. The speed of an epidemic is characterized by the exponential growth rate $r$, which measures how *fast* an epidemic grows at the population level. Knowing the strength and speed of an epidemic allows predictions about the course of the epidemic and the effectiveness of intervention strategies.

Much research has prioritized estimates of $\mathcal{R}$, and particularly its value in a fully susceptible population—called the *basic* reproductive number $\mathcal{R}_0$—because $\mathcal{R}$ has a threshold value (i.e. $\mathcal{R} = 1$) that determines whether a disease can invade, the level of equilibrium, and the effectiveness of control efforts [2,3]. The insight that an infection must, on average, cause at least one new infection in a naive population for a disease to persist goes back >100 years [4]; the idea of averaging by defining a 'typical' infection was formalized 30 years ago [3]. $\mathcal{R}$ is also of interest because it provides a *prima facie* prediction about the total *size* of an epidemic [2,5–8].

Here, we argue that the dominance of $\mathcal{R}$ over $r$ in the disease-dynamics literature is excessive: $r$ has been under-rated as a metric. Like $\mathcal{R}$, $r$ can serve as a threshold, and we show that it can also provide a useful metric for difficulty of elimination (cf. [9]). We first generalize the idea that $\mathcal{R}$ measures the difficulty of elimination by showing we can measure an intervention's 'strength' on the same scale as the reproductive number. We then show that we can likewise measure an intervention's 'speed' on the same scale as the growth rate. Thus,

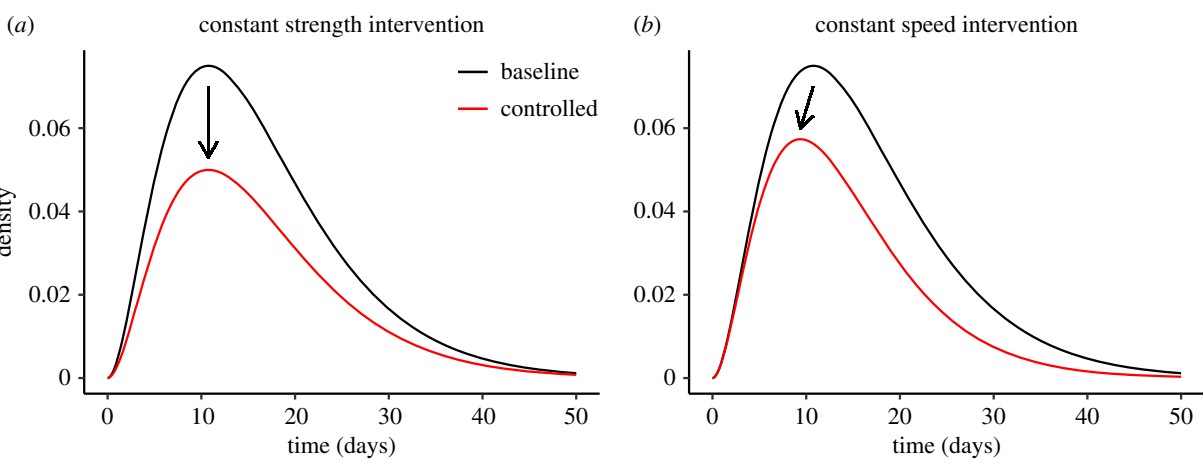

**Figure 1.** Effects of constant-strength and constant-speed intervention on infection kernels. Ebola-like gamma infection kernel (pre-intervention) $K_{pre}(\tau)$ (mean: 16.2 days, CV: 0.58 and $\mathcal{R}_{pre}$: 1.5) is shown in black [21]. The post-intervention kernel after applying each intervention strategy $K_{post}(\tau)$ is shown in red. (a) The effect of a constant-strength intervention with $\theta = 1.5$. A constant-strength intervention reduces the density by a constant proportion: $K_{post}(\tau) = K_{pre}(\tau)/\theta$; when the strength of intervention matches the strength of epidemic ($\theta = \mathcal{R}_{pre}$), the resulting distribution is equivalent to the intrinsic generation-interval distribution ($K_{post}(\tau) = g(\tau)$). (b) A constant-speed intervention with $\phi \approx 0.0267/\text{day}$ is applied to the same kernel. A constant-speed intervention reduces the density exponentially: $K_{post}(\tau) = K_{pre}(\tau) \exp(-\phi\tau)$; when the speed of intervention matches the speed of epidemic ($\phi = r_{pre}$), the resulting distribution is equivalent to the initial backward generation-interval distribution ($K_{post}(\tau) = b(\tau)$). (Online version in colour.)

there is a dual sense in which $r$ also measures difficulty of elimination. We argue that the primacy of $\mathcal{R}$ over $r$ is partly due to history, and that there are cases where speed provides the better framing for practical disease questions than strength (as well as the reverse). We provide examples of both situations, for both human immunodeficiency virus (HIV) and COVID-19.

## 2. Mathematical theory

### (a) Epidemic model

We model disease incidence using the renewal equation, a simple, flexible framework that can cover a wide range of model structures [10–16]. In our model, disease incidence at time $t$ is given by:

$$i(t) = \int K(\tau, t) i(t - \tau)\, d\tau. \tag{2.1}$$

Here, $K(\tau, t)$ is the infection kernel describing how infectious we expect an individual infected $\tau$ time units ago to be in the population. In general, $K(\tau, t)$ will depend on population characteristics that may change through time $t$—notably, the proportion of the population susceptible, $S(t)$.

When $K$ remains constant over time, the renewal equation is equivalent to the Von Foerster equations (e.g. [17]). Since we are interested in invasion and control, we will often assume that changes in $S$ through time are negligible (as would be expected when disease levels are small)—that is, $S(t) \approx S(0)$. However, this focus does not mean we are only interested in the initial period of exponential growth: the ability of a disease to spread under conditions characteristic of a naive population is commonly used as a criterion for whether it can be eliminated under general conditions [17,18].

### (b) Strength-based decomposition

If the infection kernel $K$ is not changing with time, we can write:

$$K(\tau) = \mathcal{R}g(\tau), \tag{2.2}$$

where $g(\tau)$ is the 'intrinsic' generation-interval distribution. The generation interval is defined as the time between when a person becomes infected and when that person infects another person [19]; therefore, the intrinsic generation-interval distribution $g(\tau)$ gives the relative infectiousness of an average individual as a function of time since infection [20]. Since $g$ is a distribution, it integrates to 1, and the reproductive number $\mathcal{R}$ is thus the integral of $K$.

Imagine we have a pre-intervention $K_{pre} = \mathcal{R}_{pre}g(\tau)$ (we generally suppress the suffix in $g_{pre}$ for readability) and a control measure that proportionally reduces $K$ by a factor of $\theta$, for example, by protecting a fixed fraction of susceptibles through vaccination (figure 1a). We then have:

$$K_{post}(\tau) = \left(\frac{\mathcal{R}_{pre}}{\theta}\right)g(\tau). \tag{2.3}$$

Since $g$ integrates to 1, the reduction needed to prevent invasion (or to eliminate disease) is exactly $\theta = \mathcal{R}_{pre}$. We call $\theta$ the 'strength' of the intervention; transmission is interrupted when the strength of the intervention $\theta$ is larger than the pre-intervention strength of spread $\mathcal{R}_{pre}$.

We generalize this idea by allowing an intervention strategy to reduce $K$ by different proportions over the course of an individual infection. We write the post-intervention kernel:

$$K_{post}(\tau) = K_{pre}(\tau)/L(\tau), \tag{2.4}$$

where $L(\tau)$ is the average multiplicative reduction for an individual infected time $\tau$ ago. The post-intervention reproductive number is thus:

$$\mathcal{R}_{post} = \int K_{post}(\tau)\, d\tau. \tag{2.5}$$

This framework generalizes the work of [17], who made parametric assumptions about the shape of $L(\tau)$.

We define the strength of the intervention $L$ to be $\theta = \mathcal{R}_{pre}/\mathcal{R}_{post}$. It is then straightforward to show that $\theta$ is the harmonic mean of $L(\tau)$ weighted by the intrinsic generation-interval distribution:

$$\theta = 1/\langle 1/L(\tau)\rangle_{g(\tau)}. \tag{2.6}$$

As in the constant-$L$ case above, we have that the disease cannot spread when $\theta \geq \mathcal{R}_{\text{pre}}$. In other words, $\theta$ generalizes the well-known idea that $\mathcal{R}$ is a metric for how hard a disease is to eliminate: we can measure the 'strength' of a control measure, this must exceed the strength of the disease ($\mathcal{R}$) to achieve elimination.

We note that intervention function $L$ and the strength of intervention $\theta$ need not be calculated explicitly in many contexts: they can usefully be thought of as abstractions of existing modelling practices. Modellers typically rely on mechanistic models (often based on ordinary differential equations) to model disease spread and evaluate intervention effects. By doing so, they make implicit assumptions about the shape of $L$ and therefore about $\theta$.

## (c) Speed-based decomposition

The above decomposition generalizes the argument that $\mathcal{R}$ is the key parameter in evaluating whether a disease can be controlled—one of the main foundations of historical primacy of $\mathcal{R}$. But we can in fact do an analogous decomposition based on speed, and place $r$ in a similar role.

The Euler–Lotka equation allows us to calculate the initial exponential growth rate $r$ of an epidemic given an infection kernel $K$:

$$1 = \int K(\tau) \exp(-r\tau)\, d\tau \tag{2.7}$$

By analogy with the strength-based factorization (2.2), we can rewrite (2.7) as a speed-based factorization:

$$K(\tau) = b(\tau) \exp(r\tau) \tag{2.8}$$

Like $g$, $b$ is a distribution: in this case, the initial backward generation interval, which gives the distribution of realized generation times (measured from the infectee's point of view) when the disease spreads exponentially [20,22].

Now imagine an idealized intervention that reduces transmission at a constant hazard rate $\phi$ across the disease generation (figure 1b), for example, by identifying and isolating infectious individuals. We then have:

$$K_{\text{post}}(\tau) = K_{\text{pre}}(\tau) \exp(-\phi\tau) \tag{2.9}$$

The interpretation is that average infectiousness for under this control regime is the product of the original infectiousness $K_{\text{pre}}(\tau)$ (at age of infection $\tau$) and the probability $\exp(-\phi\tau)$ of escaping the hazard of control up to that time.

Substituting (2.8):

$$K_{\text{post}}(\tau) = K_{\text{pre}}(\tau) \exp(-\phi\tau) = b(\tau) \exp((r_{\text{pre}} - \phi)\tau) \tag{2.10}$$

Since $b$ is a distribution (which integrates to 1), the reduction needed to prevent invasion (or to eliminate disease) is exactly $\phi = r_{\text{pre}}$. We call $\phi$ the 'speed' of the intervention; transmission is interrupted when the speed of the intervention is faster than the speed of spread.

We generalize this idea by allowing the hazard rate $h(\tau)$ at which $K$ is reduced to vary through time, thus:

$$K_{\text{post}}(\tau) = K_{\text{pre}}(\tau) \exp\left(-\int_0^\tau h(\sigma)\, d\sigma\right) \tag{2.11}$$

The associated post-intervention epidemic speed $r_{\text{post}}$ is given by:

$$1 = \int K_{\text{post}}(\tau) \exp(-r_{\text{post}}\tau)\, d\tau. \tag{2.12}$$

We define the speed of a general intervention to be $\phi = r_{\text{pre}} - r_{\text{post}}$. We can then show that $\phi$ is a (sort of) mean satisfying the implicit equation:

$$1 = \left\langle \frac{\exp(\phi\tau)}{\exp\left(\int_0^\tau h(\sigma)\, d\sigma\right)} \right\rangle_{b(\tau)} \tag{2.13}$$

Specifically, the speed $\phi$ is a mean of the hazard $h$ in the sense that an increase (or decrease) in $h$ produces the same sign of change in $\phi$, and if $h$ is constant across the generation then $\phi = h$. Like intervention strength $\theta$, intervention speed $\phi$ is also an abstraction—that is, the mechanistic models of interventions make implicit assumptions about the shape of the hazard rate $h$ and therefore $\phi$.

The disease cannot spread when $\phi \geq r_{\text{pre}}$. In other words, $r$, like $\mathcal{R}$, is a metric for how hard a disease is to eliminate: we can measure the 'speed' of a control measure, this must exceed the speed of the disease ($r$) to achieve elimination. Since both $\phi \geq r_{\text{pre}}$ and $\theta \geq \mathcal{R}_{\text{pre}}$ are conditions for control, they are necessarily equivalent: the speed paradigm does not provide a different answer, it provides a different way of thinking about the steps to the correct answer.

## 3. Example: HIV

In this section, we use both strength- and speed-based decompositions to evaluate intervention strategies for HIV. In particular, we study how the amount of early HIV transmission affects estimates of intervention effectiveness. These examples are not detailed estimates for specific scenarios; instead, they are meant to demonstrate how strength- and speed-based perspectives provide different qualitative insights, while yielding the same quantitative answers.

We model the pre-intervention infection kernel of the HIV as a sum of two gamma distributions:

$$K_{\text{pre}}(\tau) = \mathcal{R}_{\text{pre}}(p_{\text{early}} f_{\text{early}}(\tau) + (1 - p_{\text{early}}) f_{\text{late}}(\tau)). \tag{3.1}$$

The first component, $f_{\text{early}}(\tau)$, models early HIV transmission during the acute infection stage. We assume that $f_{\text{early}}(\tau)$ has a mean of three months [23] and a shape parameter of 3. The second component, $f_{\text{late}}$, models HIV transmission during the asymptomatic stage and the disease stage (after progression to acquired immune deficiency syndrome (AIDS)). We assume that $f_{\text{late}}(\tau)$ has a mean of 10 years [24,25] and a shape parameter of 2 (to roughly match the wide generation-interval distribution of HIV [17]). Finally, $p_{\text{early}}$ is the proportion of early HIV transmission.

The infection kernel is shown in (figure 2a) for our baseline value of $p_{\text{early}} = 0.23$. We assume that the pre-intervention speed of the epidemic is $r_{\text{pre}} = 0.452\,\text{year}^{-1}$ (figure 2b), and ask what value of $\mathcal{R}_{\text{pre}}$ would produce this rate of growth. When transmission is fast, (i.e. when $p_{\text{early}}$ is large), individuals do not need to transmit as much to achieve this speed, so the estimated value of $\mathcal{R}_{\text{pre}}$ decreases (figure 2c). Therefore, as $p_{\text{early}}$ gets smaller, we expect stronger intervention to be required in order to control the disease. Here, we are estimating the pre-intervention reproductive number $\mathcal{R}_{\text{pre}}$, using information about the initial rate of spread $r_{\text{pre}}$, as a proxy for the difficulty of eventually eliminating HIV.

We compare two different possible intervention strategies to shed light on the strength and speed decompositions. First, we consider a condom intervention that reduces HIV

*Proc. R. Soc. B* **288**: 20201556

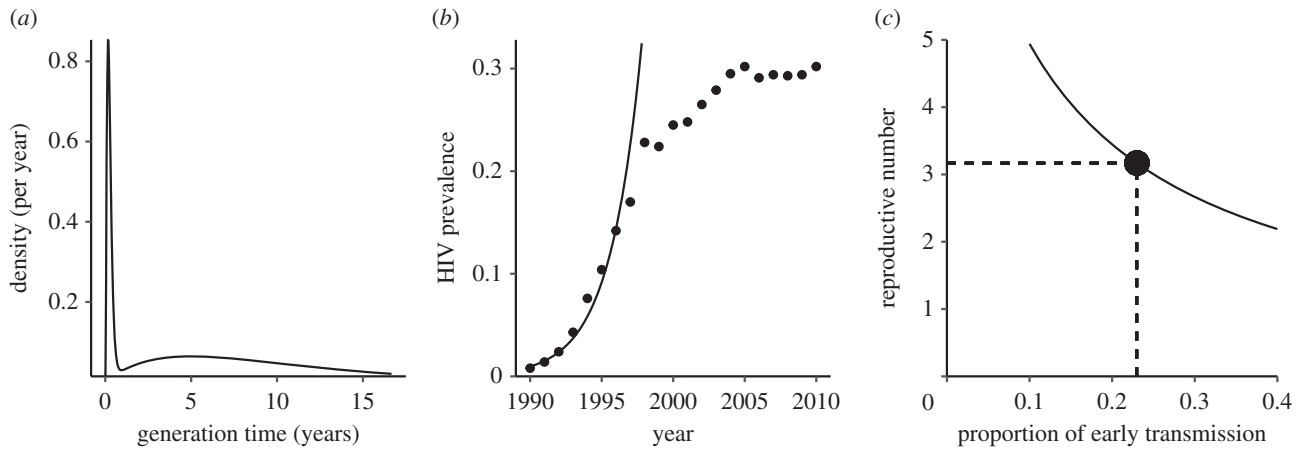

**Figure 2.** The infection kernel of HIV. (*a*) The infection kernel of HIV is approximated using a sum of two gamma distributions. We assume that the baseline proportion of early transmission is 23% [26]. (*b*) Time series of HIV prevalence in pregnant women in South Africa, 1990–2010 [27]. The initial exponential growth rate of the HIV is estimated by fitting a straight line to log-prevalence (1990–1997) by minimizing the sum of squares. (*c*) Increase in the estimate of the amount of early transmission reduces the estimate of the reproductive number. The black circle indicates the baseline scenario.

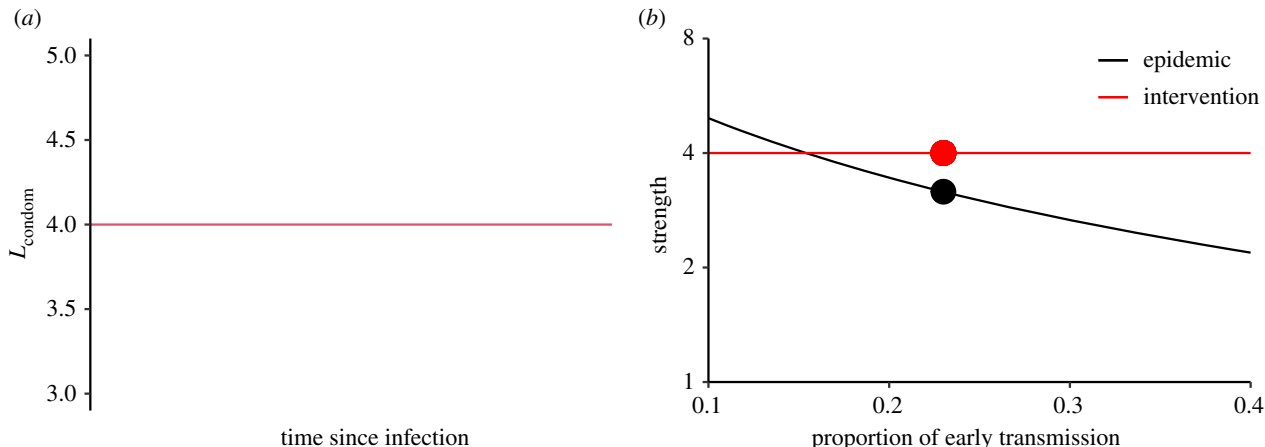

**Figure 3.** Evaluating a condom intervention using strength-based decomposition. (*a*) Condom use is thought to reduce probability of transmission by a similar factor throughout the course of infection; thus the multiplicative reduction $L_{condom}$ due to condom use is constant across the course of infection. (*b*) The estimated amount of early transmission affects estimated epidemic strength $\mathcal{R}_{pre}$, but not intervention strength $\theta$ of a condom-based intervention. The black and red circles indicate the baseline scenario. (Online version in colour.)

transmission by an average of approximately 75%. Assuming that condoms act as a physical barrier, and that condom use will, on average, remain roughly constant through time, it is reasonable to model the multiplicative reduction in transmission due to condom use as constant across the course of infection: $L_{condom} = 1/(1 - 0.75) = 4$ (figure 3*a*). The estimated strength of such an intervention is simply the average of $L_{condom}$, i.e. $\theta = 4$, whereas the estimated strength of the epidemic $\mathcal{R}_{pre}$ decreases as the proportion of early transmission $p_{early}$ increases (figure 3*b*). Thus, the predicted effectiveness of the condom intervention will depend strongly on our estimate of the importance of early transmission: if the amount of early transmission is low, we infer that disease spread is too strong to be controlled completely by our intervention.

Next, we consider a 'test-and-treat' strategy in which infected individuals are identified, linked to care and receive antiretroviral therapy (ART) with the goal of both preserving health and preventing transmission through viral suppression. [28–30]. Our assumptions for this scenario are shown in figure 4. We assume that the hazard rate $h_{test}$ of this intervention starts at 0 (because there is no way for newly infected individuals to know that they have HIV) but increases very quickly (because sexually active individuals are the most likely to seek

testing); after a few months, the assumed hazard rate goes down to account for the effects of people who avoid identification, persistent treatment failures, and the possibility of rare transmission even under effective treatment (figure 4*c*). The corresponding strength of intervention $L_{test}$ is shown in figure 4*a* and details of the assumption are given in the caption.

In this example, we see that, as $p_{early}$ goes down and our estimate of epidemic strength increases, the estimate of intervention strength increases roughly in parallel. The increase in intervention strength makes sense: less early transmission means more time to reach people before they transmit and higher strength of control. This is the core of the result of [31]. In our scenario, we predict that the intervention remains effective over the range of considered parameters.

Though there is a clear intuition for why both strengths increase as early transmission goes down, the speed paradigm provides insight into why these two increases are so close to parallel. The estimated epidemic speed depends only on the observed growth rate—it does not change if we change our assumption about the proportion of early transmission. For the test-and-treat intervention, the effective intervention speed also stays relatively constant (figure 4*d*), in part because we have (plausibly) assumed that the hazard stays relatively constant

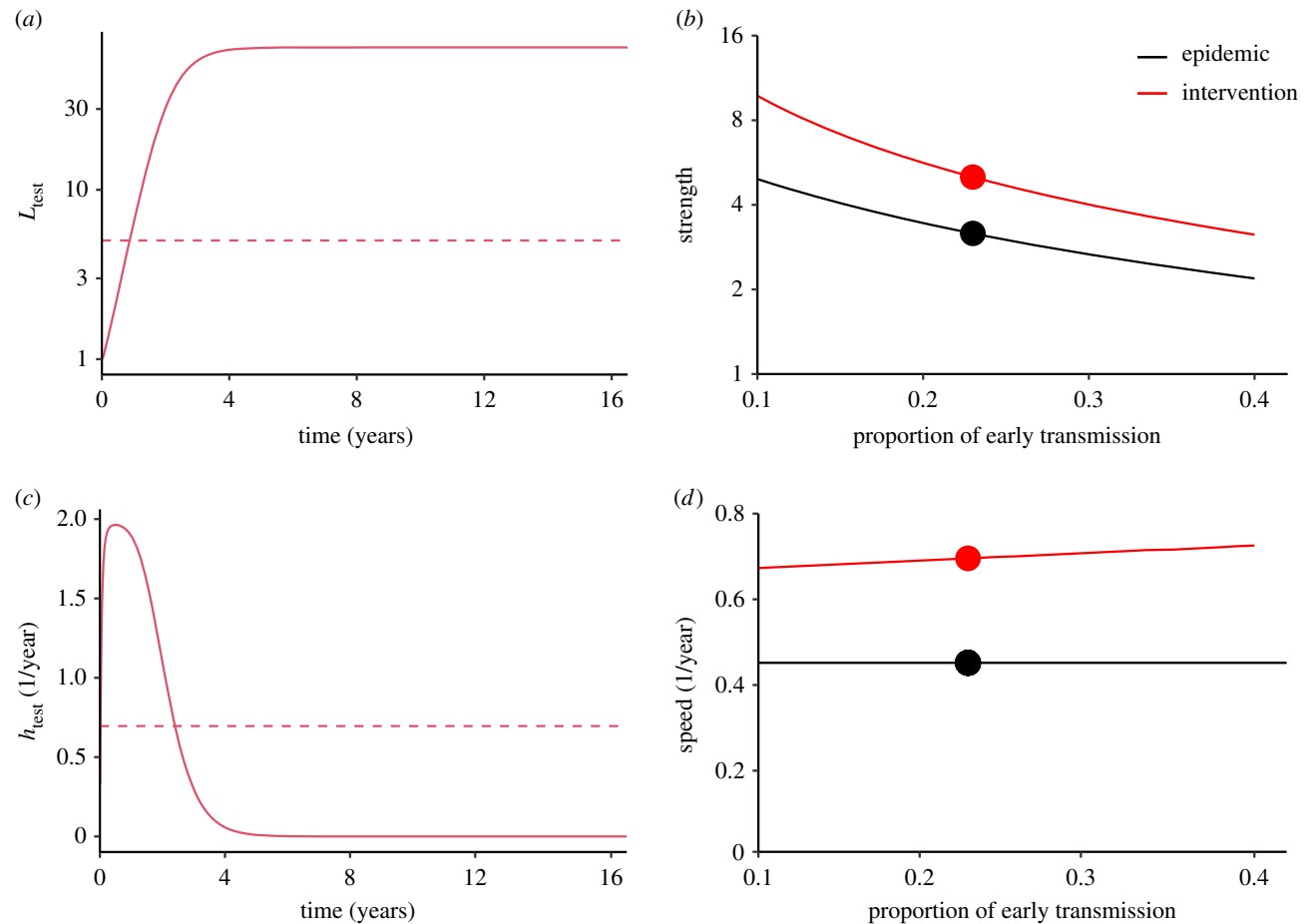

**Figure 4.** Evaluating a test-and-treat intervention using strength- and speed-based decomposition. (*a*) The strength of the test-and-treat intervention (calculated from the assumed hazard, (*c*)). The dashed line shows the corresponding effective strength of the intervention (from (2.6)) assuming 23% early transmission. (*b*) Increase in the estimated amount of early transmission decreases the estimated epidemic strength $\mathcal{R}_{pre}$ as well as the estimated strength $\theta$ of a test-and-treat intervention. (*c*) The assumed hazard for the test-and-treat intervention. The dashed line shows the corresponding effective speed of the intervention (from (2.13)) assuming 23% early transmission. (*d*) The estimated amount of early transmission has little effect on the effective speed of intervention $\phi$, and none on the pre-intervention speed $r_{pre}$ of the epidemic estimated from incidence data. Circles indicate the baseline scenario. Test-and-treat intervention is modelled phenomenologically: $L_{test}(\tau) = \exp\left(\int_0^\tau h_{test}(\sigma)\,d\sigma\right)$ and $h_{test}(\tau) = h_{max}(1 - \exp(-kf(\tau)))$, where $f(\tau)$ is a gamma probability density function with a mean of 1 year and a shape parameter of 2, $k = 4/\max(f(\tau))$, and $h_{max} = 2$ year$^{-1}$. (Online version in colour.)

for a few key months, and in part because the backward generation-interval distributions for different scenarios are relatively similar (electronic supplementary material figure). The effective intervention speed increases slightly as the proportion of early transmission increases because the subpopulation that the intervention fails to reach becomes relatively more important if late transmission is more important. Thus, the speed paradigm provides an intuitive underpinning for the originally surprising result of [31]: the effectiveness of test-and-treat interventions should not depend much on the proportion of early transmission.

We reiterate that a complete calculation using the same assumptions under either paradigm will necessarily provide consistent answers. But in this particular case, the speed paradigm provides an answer whose causes are easier to understand. We argue that it is therefore easier to assess and investigate the necessarily incomplete assumptions that underlie the conclusion.

## 4. Example: COVID-19

There has been a great deal of discussion of the importance of pre-symptomatic transmission of COVID-19 [9,32,33]. Pre-symptomatic transmission is likely to be hard to detect,

and therefore hard to prevent. Thus, it might be supposed that an increase in the estimated importance of pre-symptomatic transmission would lead to an increase in the estimated difficulty of control.

A generation-interval perspective [21] can be used to challenge this view. Here, we use a compartmental model as a concrete example (see electronic supplementary materials for details); however, the qualitative conclusions are not specific to the example. The model assumes that infected individuals progress sequentially through three stages of infection: exposed, pre-symptomatic and symptomatic. We assume that pre-symptomatic and symptomatic individuals can transmit infection at rates $\beta_p$ and $\beta_s$ for average durations of $D_p$ and $D_s$ days, respectively; thus, the pre-intervention proportion of pre-symptomatic transmission is given by:

$$p = \beta_p D_p/(\beta_p D_p + \beta_s D_s). \tag{4.1}$$

We are interested in the effect of the proportion of pre-symptomatic transmission on different intervention strategies, assuming that $r_{pre}$ is known.

More early transmission means shorter generation intervals (figure 5*a*) and, for a given value of observed $r_{pre}$, less transmission per individual—that is, a lower value of $\mathcal{R}_{pre}$ (figure 5*b*). Thus, although earlier transmission could make

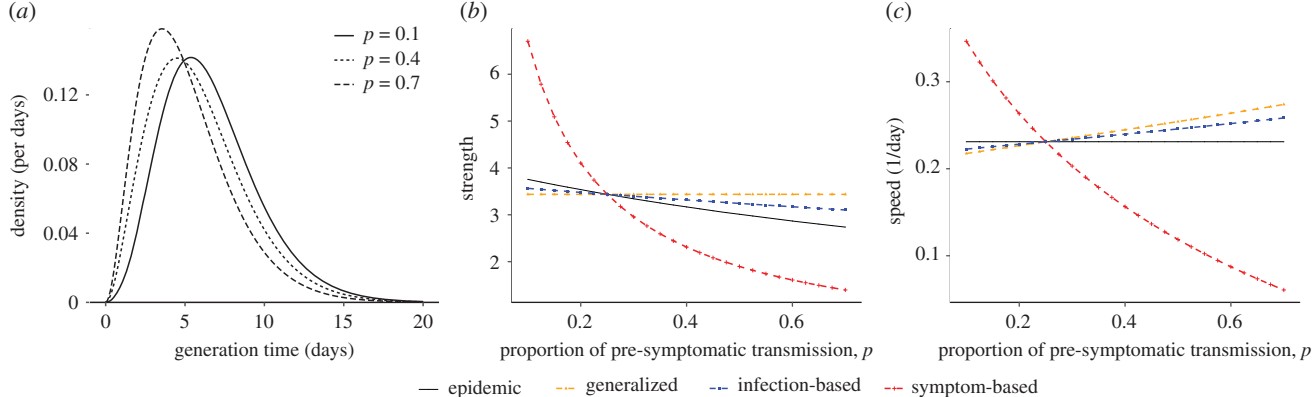

**Figure 5.** Pre-symptomatic transmission and interventions against COVID-19. (*a*) Infection kernel of COVID-19 is modelled mechanistically using gamma-distributed latent, pre-symptomatic and symptomatic stages, each with shape parameter of 2 and mean of 2.5 days. (*b*) Strength of the epidemic and of three interventions, calculated assuming a pre-intervention doubling time of 3 days (i.e. $r \approx 0.231$/day). The symptom-based intervention is modelled as a constant hazard of control for individuals in symptomatic compartments. The infection-based intervention is modelled phenomenologically (similar to the test-and-treat intervention for HIV). Control parameters were chosen so that intervention strength matches epidemic strength when $p = 0.25$. (*c*) The corresponding speeds of the epidemic and interventions. See electronic supplementary materials for details. (Online version in colour.)

intervention less effective, it also means that less intervention may be necessary. For generalized interventions like lockdowns, which are generally assumed to affect everyone roughly equally, the strength-based perspective thus gives a clear answer: more early transmission means we will conclude that non-targeted interventions are *more* effective, because the effectiveness of intervention is not affected by the amount of early transmission (figure 5*b*).

For interventions that target infected individuals, like contact tracing or test-based isolation, the speed-based paradigm provides clearer insight into the likely effects of early transmission (figure 5*c*). More early transmission does not change our estimate of the pre-intervention speed of the epidemic $r_{\text{pre}}$, which is inferred from data. But it changes our estimate of the *effective* speed of a given intervention because more early transmission gives more weight in the calculation of effective speed ($\phi$) to the early period of infection. For symptom-based interventions (e.g. self-isolation of symptomatic cases), the hazard of intervention will increase over time as individuals have higher probability of developing symptoms later on—therefore, more early transmission causes the speed of intervention to decrease and makes the control harder. For infection-based interventions (e.g. contact tracing), the hazard of intervention will start high and decrease as cases are most likely to be identified early on—in this case, more early transmission causes the effective speed of intervention to increase, and makes control easier—a result that is not obvious without the speed-based paradigm.

We also find that symptom-based interventions are extremely sensitive to the proportion of early transmission. This is because early transmission cannot, in our definition, be controlled by symptom-based interventions.

## 5. Discussion

The effectiveness of an epidemic intervention is often measured by its ability to reduce the reproductive number—$\mathcal{R}$, or outbreak 'strength'—below 1. The exponential growth rate—$r$, or outbreak 'speed'—is often seen just as a stepping stone to $\mathcal{R}$ or even overlooked entirely [34]. We argue that $\mathcal{R}$ and $r$ provide equally valid, complementary perspectives on

epidemic control, and that there are situations where each provides a clearer picture than the other.

In this study, we first extended the standard paradigm of $\mathcal{R}$ as a critical parameter for control, by defining the strength of an intervention on the same scale as $\mathcal{R}$, the strength of the epidemic (if control strength $\theta > \mathcal{R}_{\text{pre}}$ then $\mathcal{R}_{\text{post}} < 1$ and the epidemic will be controlled). We then constructed a parallel interpretation which measures the speed of an intervention on the same scale as $r$, the speed of an epidemic (if control speed $\phi > r_{\text{pre}}$ then $r_{\text{post}} < 0$ and the epidemic will be controlled). We thus showed that the standard paradigm for $\mathcal{R}$ and control has a natural parallel interpretation in terms of $r$.

To illustrate this idea, we used simple assumptions to explore the effects of two HIV intervention strategies (condoms and test-and-treat), using both strength- and speed-based frameworks. In particular, we provided an alternative explanation for the result of [31] who used detailed mathematical modelling of HIV transmission to show that the amount of early transmission has little effect on predicted effectiveness of a test and treat intervention: we can control an outbreak if we can identify infected individuals and enroll them on ART faster than the *observed* rate at which new cases are generated, which does not depend on the estimates of the amount of early transmission. The original explanation of the result relied on a strength-based argument: increasing the amount of early transmission decreases the basic reproductive number, which negatively correlates with the outcome of the ART intervention [31]. The speed paradigm provides an additional insight: since we expect more early transmission to make our estimate of intervention speed (a little) faster, higher amounts of early transmission (when controlling for the observed initial growth rate) are expected to make control via test-and-treat (a little) easier.

We also discussed the question of uncertainties introduced by the unknown proportion of COVID-19 transmission that is pre-symptomatic. We showed that the strength-based paradigm provides clear insight into how this uncertainty affects interventions targeted to the general population, while the speed-based paradigm is a clearer way to think about interventions targeted to infected people. We concluded that a higher estimate of pre-symptomatic transmission increases estimated effectiveness of contact-based interventions and decreases

estimated effectiveness of symptom-based interventions. In hindsight, these conclusions are consistent with common sense, but in practice the speed-based framework provides a clear way to think about these questions.

While both strength- and speed-based frameworks can give the same conclusion about the outcome of an intervention, sometimes one provides a clearer understanding of a given measure, as we've argued above. In general, we expect the speed-based framework to be clearer for characterizing newly invading pathogens: when an epidemic is growing exponentially, $r$ can be directly observed from case data but the reproductive number cannot be estimated with confidence [35], especially when there is large uncertainty in the shape of the generation-interval distribution [34]. Conversely, we expect the strength-based framework to be clearer for evaluating established pathogens (based on the effective proportion of the population susceptible).

Thinking explicitly about the two perspectives can also reduce confusion. Because of the dominance of the strength paradigm, researchers often explore different scenarios while holding $\mathcal{R}$ fixed. Fixed $\mathcal{R}$ is in fact a good default assumption for many endemic diseases. For invading diseases, however, $r$ is likely to be better constrained by data than $\mathcal{R}$. In this case, comparing scenarios while holding $\mathcal{R}$ fixed creates a bias that makes scenarios with faster transmission at the individual level (i.e. higher proportion of early transmission) look relatively more dangerous, because these scenarios will have $r$ faster than the observed value [21,31,36].

For interventions, we expect the speed-based framework to be clearer for evaluating intervention strategies that target infected people, like test-and-treat for HIV [29], or contact-tracing and quarantine for COVID-19 [33]. We expect the strength-based framework to be clearer for intervention strategies that target the general population, like condom use, or susceptible people, like prophylaxis. In other cases, such as real-time rollout of vaccines during an outbreak, both strength and speed approaches might be similarly uncertain because the result depends both on the speed of the rollout and the (strength-like) final coverage [37].

When comparing proposed interventions with estimated epidemic parameters to evaluate strategies, the situation is similar. Some scenarios lend themselves naturally to a single approach. For example, in the classic case of vaccination to eliminate a previously established childhood disease, both disease spread and intervention can be clearly characterized using strength [18]. In our HIV example, both the HIV epidemic and the test-and-treat intervention can be best characterized using speed. Other cases, such as using social distancing (a strength-like intervention) in the early stages of COVID-19 (epidemic speed is observed) may not fit so neatly into either paradigm, however. With sufficiently detailed assumptions, we could do a correct calculation for any scenario in any paradigm. But in many of these examples, using the more appropriate paradigm for each scenario lets us know what to expect, and may strengthen our intuition for *how* the assumptions lead to the result.

In population ecology, the duality between strength and speed is more widely recognized. For example, when a population is regulated by density dependence that affects all individuals identically, $r$ may be the best measure of fitness, but when regulation primarily affects juvenile mortality, $\mathcal{R}$ is likely to be superior [38,39]. There is also a link between the duality of these perspectives and the evolutionary trade-off between speed and strength, commonly theorized as a trade-off between $r$ and carrying capacity $K$ [40].

The importance of speed-based perspectives are still rarely recognized in the case of infectious disease, however. Responses to the 2014 Ebola outbreak in West Africa and the recent COVID-19 outbreak show an over-emphasis on strength at the expense of speed: during the early phases of both outbreaks, many disease modellers tried to estimate $\mathcal{R}_0$ but overlooked $r$. For example, only one out of seven preliminary analyses of the COVID-19 outbreak that were published as pre-prints between 23 and 26 January 2020 reported the doubling time of an epidemic [41–47]. Subsequent studies then relied on strength-based frameworks to evaluate the efficacy of speed-like interventions, such as contact tracing [33,48,49], with a few exceptions [9]. We suggest that infectious disease modellers should be aware of the complementarity of these two frameworks when analysing disease outbreaks.

Data accessibility. All code and data required to reproduce this paper available at https://github.com/mac-theobio/Speed_and_strength

Authors' contributions. J.D. did the initial analyses. S.W.P. wrote the first draft. Both authors wrote code, analysed data and revised the manuscript. Both authors approved the final version of the manuscript.

Competing interests. We declare we have no competing interests.

Funding. This work was supported in part by a grant to J.D. from the Canadian Institutes for Health Research.

Acknowledgement. J.D. thanks Elia Portnoy for discussions about intervention speed.

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
