## [Peer Review File · Proceedings of the Royal Society B: Biological Sciences]

Review History

RSPB-2020-1556.R0 (Original submission)

Review form: Reviewer 1

Recommendation

Accept as is

Scientific importance: Is the manuscript an original and important contribution to its field?

Excellent

General interest: Is the paper of sufficient general interest?

Good

Quality of the paper: Is the overall quality of the paper suitable?

Excellent

Is the length of the paper justified?

Yes

Should the paper be seen by a specialist statistical reviewer?

No

Do you have any concerns about statistical analyses in this paper? If so, please specify them explicitly in your report.

No

It is a condition of publication that authors make their supporting data, code and materials available - either as supplementary material or hosted in an external repository. Please rate, if applicable, the supporting data on the following criteria.

Is it accessible?

N/A

Is it clear?

N/A

Is it adequate?

N/A

Do you have any ethical concerns with this paper?

No

Comments to the Author

One additional reference that might be worth mentioning is O. Diekmann, H. Heesterbeek, and H. Metz, The legacy of Kermack and McKendrick, in Epidemic Models, Their Structure and Relation to Data (D. Mollinson, ed.) (1995), pp.95-115.

Review form: Reviewer 2

Recommendation

Major revision is needed (please make suggestions in comments)

Scientific importance: Is the manuscript an original and important contribution to its field?

Good

General interest: Is the paper of sufficient general interest?

Acceptable

Quality of the paper: Is the overall quality of the paper suitable?

Good

Is the length of the paper justified?

Yes

Should the paper be seen by a specialist statistical reviewer?

No

Do you have any concerns about statistical analyses in this paper? If so, please specify them explicitly in your report.

Yes

It is a condition of publication that authors make their supporting data, code and materials available - either as supplementary material or hosted in an external repository. Please rate, if applicable, the supporting data on the following criteria.

Is it accessible?

N/A

Is it clear?

N/A

Is it adequate?

N/A

Do you have any ethical concerns with this paper?

No

Comments to the Author

In this work the authors propose that “strength” and “speed” are complementary frameworks for understanding interventions in infectious disease models. This is an interesting and appealing idea based on simple re-arrangements of well-known equations for exponentially growing epidemics. As the authors themselves note, comparing growth rates r rather than reproductive numbers R is already well known in ecology. In my view this paper could bring these ideas to light in infectious disease more thoroughly and compellingly with an expanded application section, a section on the relationship to data and/or estimation from data, improved and expanded section on interventions and the distinct views provided by (and usefulness of) the two distinct frameworks, and/or with some extension of the results to population dynamics that differ from constant rate exponential growth or decay. As it stands it feels quite minimal, though I appreciate the appeal of the conceptual point.

Near equation (9) the authors need to make a more explicit link with survival analysis so that we are clear on why we multiply K by survival functions. Survival functions multiply in the sense that if there are two hazards at rates λ_1 and λ_2 then the survival function is $S(t) = S_1(t) S_2(t)$. But $K(\tau)$ is proportional to the density $g(\tau)$, and is not a survival function. What is the interpretation of the density times the survival function? Why would the interventions not change the density directly, so that g becomes \hat{g} (and then this would impact the hazard rate and survival function via the usual relationship $S(t) = \int_t^\infty (\hat{g}(x) dx)$) ?

In practice what is the need for equations (6) and (12)? Both essentially re-write the other equations; are these averages that can be linked to estimates from data, for example?

Presumably both frameworks give a decline when the growth rate is negative and growth when it is positive. Therefore they would never contradict each other as to whether an intervention was sufficient or not - is this correct? This should be made explicit. It would also be good to more explicitly state and discuss what the differences are. I realize that this is in the discussion and in some sense in Figure 4, but I found the discussion section vague, particularly since the analysis is only relevant for the case of exponential growth. In particular, where would the two frameworks give different estimates of the uncertainty surrounding the benefits of an intervention, or the comparison between two interventions? On a minor note, since condoms are ultimately an individual choice applied to individual (potential) transmission events, why are they a population-level intervention whereas test and treat is deemed individual? More details on why the two frameworks give different intuition or analytical capacity would be good.

In Figures 3 and 4, the black lines are not clear to me - the strength is defined as R/\hat{R} and the speed as $\hat{r} - r$. So what is the strength (or speed) of “the epidemic” (black lines) - what’s \hat{R} and what’s R ? It might be good to have some prevalence curves

Minor quibble: equation (4) has L described in words as the “average proportional reduction” but where the reduction is 75%, in fact the denominator is $4(1 / (1-0.75))$ -- rephrase.

Decision letter (RSPB-2020-1556.R0)

06-Aug-2020

Dear Professor Dushoff:

Your manuscript has now been peer reviewed and the reviews have been assessed by an Associate Editor. The reviewers' comments (not including confidential comments to the Editor) and the comments from the Associate Editor are included at the end of this email for your reference. As you will see, the reviewers and the Editors have raised some concerns with your manuscript and we would like to invite you to revise your manuscript to address them.

Research ethics:

Use of animals and field studies:

It is a condition of publication that you make available the data and research materials supporting the results in the article. Please see our Data Sharing Policies (<https://royalsociety.org/journals/authors/author-guidelines/#data>). Datasets should be deposited in an appropriate publicly available repository and details of the associated accession number, link or DOI to the datasets must be included in the Data Accessibility section of the

article (<https://royalsociety.org/journals/ethics-policies/data-sharing-mining/>). Reference(s) to datasets should also be included in the reference list of the article with DOIs (where available).

Please submit a copy of your revised paper within three weeks. If we do not hear from you within this time your manuscript will be rejected. If you are unable to meet this deadline please let us know as soon as possible, as we may be able to grant a short extension.

Best wishes,
Professor Gary Carvalho
mailto: proceedingsb@royalsociety.org

Associate Editor

Board Member: 1

Comments to Author:

Both reviewers agreed on the value of the concepts presented in this paper. The second reviewer makes some points, specific and general, that I think could lead to extensions/revisions of the paper that would make it stronger and of more interest. I think that this paper provides important ideas for the field and look forward to seeing a revised version.

Reviewer(s)' Comments to Author:

Referee: 1

Comments to the Author(s)

. One additional reference that might be worth mentioning is O. Diekmann, H. Heesterbeek, and H. Metz, The legacy of Kermack and McKendrick, in Epidemic Models, Their Structure and Relation to Data (D. Mollinson, ed.) (1995), pp.95-115.

Referee: 2

Comments to the Author(s)

In this work the authors propose that “strength” and “speed” are complementary frameworks for understanding interventions in infectious disease models. This is an interesting and appealing idea based on simple re-arrangements of well-known equations for exponentially growing epidemics. As the authors themselves note, comparing growth rates r rather than reproductive numbers R is already well known in ecology. In my view this paper could bring these ideas to light in infectious disease more thoroughly and compellingly with an expanded application section, a section on the relationship to data and/or estimation from data, improved and expanded section on interventions and the distinct views provided by (and usefulness of) the two distinct frameworks, and/or with some extension of the results to population dynamics that differ from constant rate exponential growth or decay. As it stands it feels quite minimal, though I appreciate the appeal of the conceptual point.

Near equation (9) the authors need to make a more explicit link with survival analysis so that we are clear on why we multiply K by survival functions. Survival functions multiply in the sense that if there are two hazards at rates λ_1 and λ_2 then the survival function is $S(t) = S_1(t) S_2(t)$. But $K(\tau)$ is proportional to the density $g(\tau)$, and is not a survival function. What is the interpretation of the density times the survival function? Why would the interventions not change the density directly, so that g becomes \hat{g} (and then this would impact the hazard rate and survival function via the usual relationship $S(t) = \int_t^\infty \hat{g}(x) dx$) ?

In practice what is the need for equations (6) and (12)? Both essentially re-write the other equations; are these averages that can be linked to estimates from data, for example?

Presumably both frameworks give a decline when the growth rate is negative and growth when it is positive. Therefore they would never contradict each other as to whether an intervention was sufficient or not - is this correct? This should be made explicit. It would also be good to more explicitly state and discuss what the differences are. I realize that this is in the discussion and in some sense in Figure 4, but I found the discussion section vague, particularly since the analysis is only relevant for the case of exponential growth. In particular, where would the two frameworks give different estimates of the uncertainty surrounding the benefits of an intervention, or the comparison between two interventions? On a minor note, since condoms are ultimately an individual choice applied to individual (potential) transmission events, why are they a population-level intervention whereas test and treat is deemed individual? More details on why the two frameworks give different intuition or analytical capacity would be good.

In Figures 3 and 4, the black lines are not clear to me - the strength is defined as R/\hat{r} and the speed as $\hat{r} - r$. So what is the strength (or speed) of “the epidemic” (black lines) - what’s \hat{r} and what’s R ? It might be good to have some prevalence curves

Minor quibble: equation (4) has L described in words as the “average proportional reduction” but where the reduction is 75%, in fact the denominator is $4(1/(1-0.75))$ -- rephrase.

Author's Response to Decision Letter for (RSPB-2020-1556.R0)

See Appendix A.

RSPB-2020-1556.R1 (Revision)

Review form: Reviewer 1

Recommendation

Accept as is

Scientific importance: Is the manuscript an original and important contribution to its field?

Excellent

General interest: Is the paper of sufficient general interest?

Excellent

Quality of the paper: Is the overall quality of the paper suitable?

Excellent

Is the length of the paper justified?

Yes

Should the paper be seen by a specialist statistical reviewer?

No

Do you have any concerns about statistical analyses in this paper? If so, please specify them explicitly in your report.

No

It is a condition of publication that authors make their supporting data, code and materials available - either as supplementary material or hosted in an external repository. Please rate, if applicable, the supporting data on the following criteria.

Is it accessible?

Yes

Is it clear?

Yes

Is it adequate?

Yes

Do you have any ethical concerns with this paper?

No

Comments to the Author

This paper really extends our understanding of epidemic models

Review form: Reviewer 2

Recommendation

Accept with minor revision (please list in comments)

Scientific importance: Is the manuscript an original and important contribution to its field?

Good

General interest: Is the paper of sufficient general interest?

Good

Quality of the paper: Is the overall quality of the paper suitable?

Good

Is the length of the paper justified?

Yes

Should the paper be seen by a specialist statistical reviewer?

No

Do you have any concerns about statistical analyses in this paper? If so, please specify them explicitly in your report.

No

It is a condition of publication that authors make their supporting data, code and materials available - either as supplementary material or hosted in an external repository. Please rate, if applicable, the supporting data on the following criteria.

Is it accessible?

N/A

Is it clear?

N/A

Is it adequate?

N/A

Do you have any ethical concerns with this paper?

No

Comments to the Author

The authors have addressed most of the concerns but have not opted to include non-constant growth, estimation or the other topics, preferring instead to focus on their main conceptual point. In the midst of a pandemic, infectious disease modelers are busy, so this is understandable and the Editors need to decide whether they would have preferred the authors to expand.

The main change in this new version is the COVID-19 example, in which the authors describe in words (but do not model or justify with mathematics) how their framework helps to interpret the relationship between presymptomatic transmission and control. They say that an increase in the estimated importance of presymptomatic transmission might be expected to increase the difficulty of control, but that more early transmission for a given r means less transmission per individual, so that controls are actually easier. I think this example does not help because it is unclear for the reasons below.

Consider the example of a policy ensuring that symptomatic people self-isolate (a school or workplace might have such a policy, sending everyone with symptoms home immediately). This reduces transmission, but what transmission remains is from a - and pre-symptomatic individuals. Here, the growth rate r isn't fixed - there's less transmission. The "importance" of pre-symptomatic transmission is higher, though. So this is perhaps not an example, but what is an intervention other than something targeting symptomatic individuals that would accomplish such a shift and why would it leave r constant?

The crux of the argument is the phrase "for a given r " -- and use of the generic "importance" instead of either "amount" or "fraction" (both of which would have made clear whether the intervention reduces the total transmission by reducing transmission after symptoms, or does something else that shifts the timing of transmission). Also in this particular case, with the policy

that symptomatics go home, actually other interventions (like contact tracing) do get harder in the sense that earlier transmission is harder to stop by contact tracing, even if less of it is needed because the total number of transmissions expected has been reduced.

I think the other reason it's not quite right is that it's hard to think of an intervention that would leave the rate of growth fixed but change the timing of transmission (so changing the relationship between R and the rate of growth by changing the generation interval). So the conditioning on constant r is a little hard to picture.

In the response document there are many points where the authors note that R might not be constant but r might be, or vice versa. Concrete examples would help. I could imagine that constant R (people infect the same mean number) but different r (they do so more or less quickly) could occur if the course of viral infection within the host changes (for genetic or host reasons); if people compensate for good symptomatic testing by taking more risks before symptoms.. reaching the same R but at a different time in their infection; but these feel quite just-so and I am left with the same sense of reaching for a concrete example that I had in the last version.

Decision letter (RSPB-2020-1556.R1)

15-Dec-2020

Dear Professor Dushoff:

Your manuscript has now been peer reviewed and the reviews have been assessed by an Associate Editor. The reviewers' comments (not including confidential comments to the Editor) and the comments from the Associate Editor are included at the end of this email for your reference. As you will see, the reviewers and the Editors have raised some concerns with your manuscript and we would like to invite you to revise your manuscript to address them.

Research ethics:

Use of animals and field studies:

It is a condition of publication that you make available the data and research materials supporting the results in the article (<https://royalsociety.org/journals/authors/author-guidelines/#data>). Datasets should be deposited in an appropriate publicly available repository and details of the associated accession number, link or DOI to the datasets must be included in the Data Accessibility section of the article (<https://royalsociety.org/journals/ethics-policies/data-sharing-mining/>). Reference(s) to datasets should also be included in the reference list of the article with DOIs (where available).

Please submit a copy of your revised paper within three weeks. If we do not hear from you within this time your manuscript will be rejected. If you are unable to meet this deadline please let us know as soon as possible, as we may be able to grant a short extension.

Best wishes,
Professor Gary Carvalho

Editor, Proceedings B
 mailto: proceedingsb@royalsociety.org

Associate Editor
 Board Member: 1

Comments to Author:

The referees are generally positive about the changes that were made to the manuscript. However, one reviewer continues to raise important questions about the covid-19 example - and also about whether an example of this type is possible. I strongly recommend that these points are carefully reflected upon and new discussion is included in the next version. Overall, I think that this is a very good contribution to the epidemic modelling literature. I might not have thought that before 2020 but this pandemic has made me think differently.

Reviewer(s)' Comments to Author:

Referee: 1

Comments to the Author(s)

This paper really extends our understanding of epidemic models

Referee: 2

Comments to the Author(s)

The authors have addressed most of the concerns but have not opted to include non-constant growth, estimation or the other topics, preferring instead to focus on their main conceptual point. In the midst of a pandemic, infectious disease modelers are busy, so this is understandable and the Editors need to decide whether they would have preferred the authors to expand.

The main change in this new version is the COVID-19 example, in which the authors describe in words (but do not model or justify with mathematics) how their framework helps to interpret the relationship between presymptomatic transmission and control. They say that an increase in the estimated importance of presymptomatic transmission might be expected to increase the difficulty of control, but that more early transmission for a given r means less transmission per individual, so that controls are actually easier. I think this example does not help because it is unclear for the reasons below.

Consider the example of a policy ensuring that symptomatic people self-isolate (a school or workplace might have such a policy, sending everyone with symptoms home immediately). This reduces transmission, but what transmission remains is from a - and pre-symptomatic individuals. Here, the growth rate r isn't fixed - there's less transmission. The "importance" of pre-symptomatic transmission is higher, though. So this is perhaps not an example, but what is an intervention other than something targeting symptomatic individuals that would accomplish such a shift and why would it leave r constant?

The crux of the argument is the phrase "for a given r " -- and use of the generic "importance" instead of either "amount" or "fraction" (both of which would have made clear whether the intervention reduces the total transmission by reducing transmission after symptoms, or does something else that shifts the timing of transmission). Also in this particular case, with the policy that symptomatics go home, actually other interventions (like contact tracing) do get harder in the sense that earlier transmission is harder to stop by contact tracing, even if less of it is needed because the total number of transmissions expected has been reduced.

I think the other reason it's not quite right is that it's hard to think of an intervention that would leave the rate of growth fixed but change the timing of transmission (so changing the relationship between R and the rate of growth by changing the generation interval). So the conditioning on constant r is a little hard to picture.

In the response document there are many points where the authors note that R might not be constant but r might be, or vice versa. Concrete examples would help. I could imagine that constant R (people infect the same mean number) but different r (they do so more or less quickly) could occur if the course of viral infection within the host changes (for genetic or host reasons); if people compensate for good symptomatic testing by taking more risks before symptoms..

reaching the same R but at a different time in their infection; but these feel quite just-so and I am left with the same sense of reaching for a concrete example that I had in the last version.

Author's Response to Decision Letter for (RSPB-2020-1556.R1)

See Appendix B.

Decision letter (RSPB-2020-1556.R2)

24-Feb-2021

Dear Professor Dushoff

I am pleased to inform you that your manuscript entitled "Speed and strength of an epidemic intervention" has been accepted for publication in Proceedings B.

Open Access

Your article has been estimated as being 7 pages long. Our Production Office will be able to confirm the exact length at proof stage.

Paper charges

Sincerely,
Professor Gary Carvalho
Editor, Proceedings B
mailto: proceedingsb@royalsociety.org

Associate Editor:
Board Member
Comments to Author:
(There are no comments.)

Appendix A

Dear Editor:

Thank you for the chance to revise and resubmit our manuscript. We have made major revisions to our manuscript to address the reviewers' comments. We have also added a new section and discuss how presymptomatic transmission can affect the efficacy of COVID-19 interventions. Below please find our detailed responses to reviewers.

Reviewer #1

One additional reference that might be worth mentioning is O. Diekmann, H. Heesterbeek, and H. Metz, The legacy of Kermack and McKendrick, in Epidemic Models, Their Structure and Relation to Data (D. Mollinson, ed.) (1995), pp.95-115.

Thank you. We now cite this article as an early mentioner of the little-r threshold.

Reviewer #2

In this work the authors propose that “strength” and “speed” are complementary frameworks for understanding interventions in infectious disease models. This is an interesting and appealing idea based on simple re-arrangements of well-known equations for exponentially growing epidemics. As the authors themselves note, comparing growth rates r rather than reproductive numbers R is already well known in ecology.

In my view this paper could bring these ideas to light in infectious disease more thoroughly and compellingly with an expanded application section, a section on the relationship to data and/or estimation from data, improved and expanded section on interventions and the distinct views provided by (and usefulness of) the two distinct frameworks, and/or with some extension of the results to population dynamics that differ from constant rate exponential growth or decay. As it stands it feels quite minimal, though I appreciate the appeal of the conceptual point.

Thank you for this feedback. We agree that these ideas are similar to ideas that have been discussed in the ecological context, but also feel that this is a perspective sorely lacking in the disease-modeling world.

We have expanded the application section with examples from COVID-19, providing a qualitative examination of how uncertainties in the amount of presymptomatic transmission can affect conclusions about the efficacy of intervention. In the beginning of the current pandemic, there was much discussion of how presymptomatic transmission could make intervention more difficult—however, some of these conclusions were based on the assumption that \mathcal{R} is constant, rather than r (e.g., Hellewell et al. 2020. “Feasibility of controlling COVID-19 outbreaks by isolation of cases and contacts.” *Lancet Global Health*). In reality, given observed r , more presymptomatic transmission reduces \mathcal{R} , meaning that strength-based interventions will be easier than previously thought. To our mind, these examples further underline the absence of r -based thinking in the disease-modeling world.

While we appreciate the reviewer’s concern that a more data-driven approach could provide further insight, we wanted to keep the focus on the basic general principles without getting bogged down in the details of specific model or data-fitting assumptions.

We now address more explicitly the link between the parameters of the initial phase and the prospects for long-term control. We do not go more deeply here into the subject of non-exponential disease dynamics. We are actively working on this topic, and the issues are quite complicated, so we feel it’s best to present these in another paper.

Near equation (9) the authors need to make a more explicit link with survival analysis so that we are clear on why we multiply K by survival functions. Survival functions multiply in the sense that if there are two hazards at rates λ_1 and λ_2 then the survival function is $S(t) = S_1(t) S_2(t)$. But $K(\tau)$ is proportional to the density $g(\tau)$, and is not a survival function. What is the interpretation of the density times the survival function? Why would the interventions not change the density directly, so that g becomes g_{hat} (and then this would impact the hazard rate and survival function via the usual relationship $S(t) = \int_t^\infty (\hat{g}(x) dx)$) ?

We have expanded (9) into two steps and attempted to explain the logic clearly:

“Now imagine an idealized intervention that reduces transmission at a constant hazard rate ϕ across the disease generation (Fig. 1B), for example, by identifying and isolating infectious individuals. We then have:

$$\hat{K}(\tau) = K(\tau) \exp(-\phi\tau) \quad (1)$$

The interpretation is that average infectiousness for under this control regime is the product of the original infectiousness $K(\tau)$ (at age of infection τ) and the probability $\exp(-\phi\tau)$ of escaping the hazard of control up to that time.

Substituting (8):

$$\hat{K}(\tau) = K(\tau) \exp(-\phi\tau) = b(\tau) \exp((r - \phi)\tau) \quad (2)$$

Since b is a distribution (which integrates to 1), the reduction needed to prevent invasion (or to eliminate disease) is exactly $\phi = r$. We call ϕ the “speed” of the intervention; transmission is interrupted when the speed of the intervention is faster than the speed of spread.”

In practice what is the need for equations (6) and (12)? Both essentially rewrite the other equations; are these averages that can be linked to estimates from data, for example?

We think that the value of (6) and (12 – now 13) is conceptual. We have added explanations following these derivations. In retrospect, we believe that S2.2 and S2.3 were written in too “mathematical” a style: we got a bit carried away with the formal parallelism. The new version is still short, but engages more with the biological concepts.

We note that in practice 6 and 12 are often implicit in mechanistic models. We now make this point in the MS near the end of both S2.2 and S2.3:

“ We note that intervention function L and the strength of intervention θ need not be calculated explicitly in many contexts: they can usefully be thought of as abstractions of existing modeling practices. Modelers typically rely on mechanistic models (often based on ordinary differential equations)

to model disease spread and evaluate intervention effects. By doing so, they make implicit assumptions about the shape of L and therefore θ . ”

“Like intervention strength θ , intervention speed ϕ is also an abstraction — that is, the mechanistic models of interventions make implicit assumptions about the shape of the hazard rate h and therefore ϕ .”

Presumably both frameworks give a decline when the growth rate is negative and growth when it is positive. Therefore they would never contradict each other as to whether an intervention was sufficient or not - is this correct? This should be made explicit.

This is correct, and we have worked to be more explicit, specifically at the end of S2 and the end of S3.

It would also be good to more explicitly state and discuss what the differences are. I realize that this is in the discussion and in some sense in Figure 4, but I found the discussion section vague, particularly since the analysis is only relevant for the case of exponential growth. In particular, where would the two frameworks give different estimates of the uncertainty surrounding the benefits of an intervention, or the comparison between two interventions?

We never expect to get a different final answer from the two formalisms if they are both correctly applied; they are two ways of doing essentially the same calculation. We have tried to be more clear about this as mentioned above. We do expect in some cases to be able to use one formalism or another to more clearly illuminate the key issues and quantities determining the detailed answer, as we try to explain in our HIV example and in our new COVID example.

Keeping the two paradigms in mind, however, can help to do the analysis correctly. In particular, in emerging epidemics, researchers often vary assumptions about parameters while holding \mathcal{R} constant, even in cases where it would be more consistent with available information to hold r constant. We now discuss this in the paper:

Thinking explicitly about the two perspectives can also reduce confusion. Because of the dominance of the strength paradigm, researchers often explore

different scenarios while holding \mathcal{R} fixed. Fixed \mathcal{R} is in fact a good default assumption for many endemic diseases. For invading diseases, however, r is likely to be better constrained by data than \mathcal{R} . In this case, comparing scenarios while holding \mathcal{R} fixed creates a bias that makes scenarios with faster transmission at the individual level (i.e., higher proportion of early transmission) look relatively more dangerous, because these scenarios will have r faster than the observed value . . .

On a minor note, since condoms are ultimately an individual choice applied to individual (potential) transmission events, why are they a population-level intervention whereas test and treat is deemed individual? More details on why the two frameworks give different intuition or analytical capacity would be good.

Thank you for this. We were using individual to mean targeted at infectious individuals, and population to mean targeted at everyone. We have now dropped these shortcuts, and we simply explain this directly:

“We expect the strength-based framework to be clearer for intervention strategies that target the general population, like condom use, or susceptible people, like prophylaxis.”

We have also added more discussion of how frameworks differ, both in examples and in Discussion.

In Figures 3 and 4, the black lines are not clear to me – the strength is defined as R/\hat{R} and the speed as $\hat{r} - r$. So what is the strength (or speed) of “the epidemic” (black lines) - what’s \hat{R} and what’s R ? It might be good to have some prevalence curves

We define strength and speed of the epidemic in the introduction, and compare these to strength and speed of intervention efforts. We now reiterate these points (including in the figure captions).

Minor quibble: equation (4) has L described in words as the “average proportional reduction” but where the reduction is 75%, in fact the denominator is 4: $(1/(1 - 0.75))$ – rephrase.

Done, thank you.

Appendix B

The referees are generally positive about the changes that were made to the manuscript. However, one reviewer continues to raise important questions about the covid-19 example - and also about whether an example of this type is possible. I strongly recommend that these points are carefully reflected upon and new discussion is included in the next version. Overall, I think that this is a very good contribution to the epidemic modelling literature. I might not have thought that before 2020 but this pandemic has made me think differently.

Thank you for your help with this MS, and for your willingness to help us find a middle ground. We have maintained our focus on counterfactuals involving known, and roughly constant, values of r and R , but now explain this a little better. Crucially, we have changed the way we talk about our counterfactuals, in a way we think will be helpful to readers like Referee 2. We are now much more careful to clarify when we are talking about our pre-intervention assumptions, and we avoid a former confusion between generic values and pre-intervention values by using an explicit subscript “pre”.

Referee: 1

This paper really extends our understanding of epidemic models

Thank you for your help with this MS, and for your positive words.

Referee: 2

Thank you for your continued patience and work to help us improve this MS.

The authors have addressed most of the concerns but have not opted to include non-constant growth, estimation or the other topics, preferring instead to focus on their main conceptual point. in the midst of a pandemic, infectious disease modelers are busy, so this is understandable and the Editors need to decide whether they would have preferred the authors to expand.

The main change in this new version is the COVID-19 example, in which the authors describe in words (but do not model or justify with mathematics) how their framework helps to interpret the relationship between presymptomatic transmission and control. They say that an increase in the estimated importance of presymptomatic transmission might be expected to increase the difficulty of control, but that more early transmission for a given r means less transmission per individual, so that controls are actually easier. I think this example does not help because it is unclear for the reasons below.

Consider the example of a policy ensuring that symptomatic people self-isolate (a school or workplace might have such a policy, sending everyone with symptoms home immediately). This reduces transmission, but what transmission remains is from a- and pre-symptomatic individuals. Here, the growth rate r isn't fixed - there's less transmission. The "importance" of pre-symptomatic transmission is higher, though. So this is perhaps not an example, but what is an intervention other than something targeting symptomatic individuals that would accomplish such a shift and why would it leave r constant?.

The crux of the argument is the phrase "for a given r " – and use of the generic "importance" instead of either "amount" or "fraction" (both of which would have made clear whether the intervention reduces the total transmission by reducing transmission after symptoms, or does something else that shifts the timing of transmission). Also in this particular case, with the policy that symptomatics go home, actually other interventions (like contact tracing) do get harder in the sense that earlier transmission is harder to stop by contact tracing, even if less of it is needed because the total number of transmissions expected has been reduced.

I think the other reason it's not quite right is that it's hard to think of an intervention that would leave the rate of growth fixed but change the timing of transmission (so changing the relationship between R and the rate of growth by changing the generation interval). So the conditioning on constant r is a little hard to picture.

We feel this is a problem with our explanation, not with our argument. We have broadened our attempts to explain our conceptual framework (see below). In particular, in no case do we imagine an intervention that leaves r fixed: what we are fixing is the the *pre-intervention* value of r in our two examples. The idea is that r_{pre} has been observed more clearly than R or the generation interval for HIV and for COVID-19, so when we change our estimate of g_{pre} (called g in the MS), it changes our estimate of R_{pre} while leaving r_{pre} fixed.

In the response document there are many points where the authors note that R might not be constant but r might be, or vice versa. Concrete examples would help. I could imagine that constant R (people infect the same mean number) but different r (they do so more or less quickly) could occur if the course of viral infection within the host changes (for genetic or host reasons) ; if people compensate for good symptomatic testing by taking more risks before symptoms.. reaching the same R but at a different time in their infection; but these feel quite just-so and I am left with the same sense of reaching for a concrete example that I had in the last version.

These comments are very useful in underlining how we've explained poorly. The issues about what counter-factuals to account for in thinking about "initial" values of r and R are complicated, and we previously tried to avoid them. This made our submissions harder to follow than they should have been.

We now systematically use the subscripts pre and post for both r and R , and have tried to make clear that it is r_{pre} that we are assuming is known in the examples discussed above. We believe that this has made our paper substantially easier to read, and thank the reviewer for the patience that it took to help us see that we needed to be more clear.